# The Tumor Microenvironment and the Estrogen Loop in Thyroid Cancer

**DOI:** 10.3390/cancers15092458

**Published:** 2023-04-25

**Authors:** Nerina Denaro, Rebecca Romanò, Salvatore Alfieri, Alessia Dolci, Lisa Licitra, Imperia Nuzzolese, Michele Ghidini, Claudia Bareggi, Valentina Bertaglia, Cinzia Solinas, Ornella Garrone

**Affiliations:** 1Medical Oncology, Fondazione Istituto di Ricovero e Cura a Carattere Scientifico (IRCCS) Ca’ Granda Ospedale Maggiore Policlinico, 20122 Milan, Italy; 2Head and Neck Medical Oncology Department, Fondazione IRCCS Istituto Nazionale dei Tumori, 20133 Milan, Italy; 3Endocrinology Unit, Fondazione Istituto di Ricovero e Cura a Carattere Scientifico (IRCCS) Ca’ Granda Ospedale Maggiore Policlinico, 20122 Milan, Italy; 4Department of Hematology and Oncology, University of Milan, 20122 Milan, Italy; 5Department of Oncology, University of Turin, S. Luigi Gonzaga Hospital, 10043 Orbassano, Italy; 6Medical Oncology, AOU Cagliari, Policlinico di Monserrato, 09042 Cagliari, Italy

**Keywords:** thyroid cancer, estrogens, estrogen receptor, sex hormones, tumor microenvironment

## Abstract

**Simple Summary:**

The composite network of stromal, immune, vascular and cancer cells, which constitutes the tumor microenvironment (TME), has emerged as a player in thyroid cancer (TC) development and progression, as well as in several other cancer settings. In this context, estrogens may also contribute to TC carcinogenesis by activating proliferative pathways (namely PI3K/AKT/mTOR and RAS/Raf/MAPK), as well as by exerting immunosuppressive, pro-carcinogenic effects. To date, the complex interactions between the TME and estrogenic pathways have not been entirely unraveled in TC.

**Abstract:**

Thyroid cancer (TC) cells employ multiple signaling pathways, such as PI3K/AKT/mTOR and RAS/Raf/MAPK, fostering cell proliferation, survival and metastasis. Through a complex interplay with immune cells, inflammatory mediators and stroma, TC cells support an immunosuppressive, inflamed, pro-carcinogenic TME. Moreover, the participation of estrogens in TC pathogenesis has previously been hypothesized, in view of the higher TC incidence observed among females. In this respect, the interactions between estrogens and the TME in TC could represent a relevant, unexplored area of research. We thereby collectively reviewed the available evidence concerning the potential carcinogenic role of estrogens in TC, specifically focusing on their crosstalk with the TME.

## 1. Introduction

Thyroid cancer (TC) incidence has been steadily increasing in the last decades worldwide [1]. Differentiated thyroid carcinoma (DTC), which derives from follicular cells, is the most common form, comprising papillary thyroid carcinoma (PTC), follicular thyroid carcinoma (FTC), poorly differentiated carcinoma (PDTC) and other rare histotypes. On the other hand, anaplastic thyroid cancer (ATC) is a highly aggressive form accounting for about 2% of all thyroid malignancies, while medullary thyroid cancer (MTC), which derives from parafollicular C cells, accounts for 5% of all TC cases [2] (Figure 1).

Across the last decade, the growing interest and knowledge concerning TC’s molecular pathogenesis has led to encouraging therapeutic advancements. First, the approval of tyrosine kinase inhibitors (TKIs)—namely lenvatinib, sorafenib and cabozantinib—showed a positive impact on the clinical outcomes of radioiodine refractory (RAIR) DTC patients. These were followed by the development of selective RET inhibitors (selpecartinib and pralsetinib) and tropomyosin receptor kinase (TRK) inhibitors (larotrectinib and entrectinib) for the advanced/metastatic DTC harboring RET fusion gene and TRK fusion gene, respectively. Furthermore, RET inhibitors and the combination of BRAF-MEK inhibitors gained a role in the management of MTC and ATC, respectively [2].

Despite this enhanced therapeutic armamentarium and the implemented techniques of secondary prevention, TC mortality rates are still far from a substantial improvement [3].

To date, several genetic alterations are known to be involved in TC development. These include transverse point mutations of BRAF yielding the altered BRAFV600E protein, as well as mutations involving RAS, PTEN-phosphatidylinositol-3 kinase (PI3K)-AKT pathway), B catenin, TP53 and isocitrate dehydrogenase1 (IDH1).

Copy amplifications often involve PI3K-AKT genes, while the most frequent translocations are represented by RET-PTC and paired box 8 (PAX8)-peroxisome proliferator-activated receptor (PPARG); epigenetic alterations have also been reported. In TC, the *Wnt* pathway is also implicated in stem cells’ maintenance [4].

As far as DTC and ATC are concerned, the main mutations responsible for disease progression and aggressiveness include TP53-inactivating mutations, activating mutations along the *Wnt*/β-catenin pathway and activating mutations of the telomerase reverse transcriptase (TERT) promoter. Dysregulation of the mitogen-activated protein kinase (MAPK) and PI3K/AKT signaling pathways also play a leading role in this context. On the other hand, mutations in the *RET* proto-oncogene represent the primary molecular drivers of MTC tumorigenesis [4].

Moving beyond the genetic pathways strictly related to thyroid follicular cells, the focus of clinical research has more recently shifted towards the role of the tumor microenvironment (TME), which, according to its characteristics, was shown to have a variable impact on oncologic outcomes across several solid tumors [5,6]. The TME can be defined as a complex network of malignant cells, non-immune cells (endothelial cells, stromal cells and cancer-associated fibroblasts (CAFs)) and a variety of immune cells (macrophages, polymorphonuclear cells, mast cells, natural killer (NK) cells, dendritic cells (DCs), and T and B lymphocytes), which play a central role in tumor growth and invasion.

In RAIR-TC patients not responding to TKIs, an inflamed, immunosuppressive TME has been described. In this context, TC cells produce cytokines and chemokines that cause an imbalance between T effectors and regulators and attract tumor-associated macrophages (TAMs) and myeloid-derived suppressor cells (MDSCs) [5]: such complex interplay, involving both pro- and anti-tumoral molecules, affects the overall ratio between immune effector and immune-suppressive cells, with an impact on the processes of TC cells’ proliferation and survival.

The participation of estrogens in TC pathogenesis has also previously been hypothesized: indeed, TC (as well as benign thyroid diseases, in particular autoimmune conditions) shows a female preponderance, with DTC displaying a higher female/male ratio as compared to PDTC/ATC [7]. While the implications of estrogens in autoimmunity have been more clearly documented [8,9], evidence supporting a correlation between thyroid autoimmune disease and TC is historically controversial: as of today, most of the available data seem to point towards a supposedly protective role of thyroid autoimmunity against TC [10,11,12].

Among estrogens, estradiol (E2) is the most potent ligand with the highest affinity. Interestingly, estrogens have been shown to exert their activity through both classical (i.e., genomic) and non-genomic signaling.

Estrogen receptors (ERs) modulate the PI3K/AKT/mTOR and RAS/RAF/MEK pathways, stimulate reactive oxygen species (ROS) production and promote cell cycle proliferation during the G1-S phase through the modulation of Cyclin D1 [13] (Figure 2). Within the TME in TC, ERs are largely expressed on the extracellular membrane of CAFs, TAMs and MDSCs.

In this context, an interest concerning female sex hormones and their crosstalk with the TME has emerged, driving a promising line of research in the TC setting. In this review, we report the currently available knowledge on the role of the TME and estrogens in TC pathogenesis and development, especially focusing on the possible, mutual interactions between these two players.

## 2. Materials and Methods

An electronic literature search was conducted on the PubMed database for English articles published up to 30 December 2022. Boolean operators (OR, AND) were used to combine the following search terms: “thyroid cancer”, “tumor microenvironment”, “TME”, “estrogen”, “estrogen receptor”, “molecular pathogenesis”.

Three independent reviewers (N.D., S.A and I.N.) screened titles and abstracts and performed the final article selection. Any discrepancy was resolved by discussion with the other reviewers. Thereafter, meeting proceedings (European Society of Medical Oncology—ESMO, European Thyroid Association—ETA, American Society of Clinical Oncology—ASCO and American Thyroid Association—ATA), reference lists of published studies, review articles and relevant books were also considered. RR revised the paper editing. The Results section provides an overview on the role of TME and estrogens, as well as on their mutual interactions, in TC setting.

## 3. Results

### 3.1. TME in TC: Immune Cells, Inflammatory Mediators and Stroma

The TME immune infiltrate specifically varies among the different TC subtypes: in DTC, a higher number of tumor-associated lymphocytes and T regulatory cells (Tregs) is found, while ATC and MTC display a higher density of TAMs. Evidence suggesting the potential pro-tumorigenic function of an inflamed TME has been collected concerning PTC [5]. Specifically, Fugazzola et al. revealed that the expression of three inflammation-related genes (CCL20, CXCL8 and l-selectin) was enhanced in BRAFV600E and in RET/PTC tumors, as compared to normal samples: this resulted in a reciprocal interaction among tumor cells, stromal cells and all the other TME components, actively fostering an inflamed, pro-tumorigenic microenvironment [14].

Several studies also evaluated the presence of cytokines and chemokines within the TME, supporting their role in the regulation of the interlacing networks among TC immune and tumor cells. The main pro-tumoral cytokines include tumor necrosis factor (TNF)-α, transforming growth factor (TGF)-α, IL6, IL4, IL10 and IL17, while, among anti-tumoral molecules, interferon (IFN)-α, IL2 and TGF-β play a significant role; in this setting, it is relevant to underline that most cytokines display a pleiotropic activity, with an effect both on innate and on adaptive immune mechanisms [15].

Specifically, Wen et al. evaluated a prognostic signature including CAFs score, VEGF, TNF-α, IL2/STAT5 and IL6/JAK/STAT3, where the aberrant activation of the RAS/RAF MAPK and PI3K/AKT pathways was shown to be implicated in TC progression [16]. The IL6/JAK/STAT3 pathway also correlated with poor TC outcomes, through the maintenance of an inflamed, immunosuppressive TME: indeed, constitutively active STAT3 increases the production of VEGF, IL10 and TGF-α, thereby impairing cytotoxic T cells’ function while promoting TAMs’ and MDSCs’ proliferation; these, in return, produce reactive oxygen and nitrogen species, further supporting a pro-carcinogenic, inflamed TME [16,17].

High levels of Tregs, monocytes, IL10 and chemokines (such as CCL12, CCL2) were also shown to associate with a higher risk of TC lymph node metastases and overall worse prognosis [18,19,20]. Furthermore, Kim et al. demonstrated that the number of TAMs within TME was proportionally correlated with TC primary tumor size [21].

In this scenario, the prevalence of programmed death 1 ligand (PD-L1) expression in TC has not been well-established. PD-L1 positivity in DTC ranges from 6% to 87.5%, depending on different antibody assays. Moreover, PD-L1 expression varies when evaluated by the tumor positive score (TPS) or combined positive score (CPS), with the latter often providing higher values. A higher CPS has been correlated with worse TC prognosis; furthermore, in the case of the same PD-L1 value being reported for the TPS and the CPS, the latter was characterized by poorer outcomes: in other words, the negative impact of the stromal TME on TC outcomes is better described by the CPS score, as compared to the TPS score [22]. Notably, TC cell lines with BRAFV600E mutation display higher PD-L1 mRNA expression as compared to BRAF wild type cells [23,24]. Based on this background, several clinical trials involving oncological treatments in combination with immune checkpoint inhibitors in TC are currently ongoing (Table 1).

Among several key mechanisms of immune evasion that hamper antigen presentation in TC—including high levels of vascular endothelial growth factor (VEGF), PGE2, IL-10, TGF-β and galectin-1, and the abundance of Tregs and TAMs [25,26,27,28]—the modulation of major histocompatibility complex (MHC) expression has also emerged. In this respect, Giuliani et al. demonstrated that MHC reduction was associated with high levels of TGF-β1, and α- and γ-interferons (IFNs). Other negative MHC regulators included insulin and insulin-like growth factor receptor (IGF)-1, Thyroid Stimulating Hormone (TSH), methimazole, phenyl methimazole, thymosin-α1 and glucose [29].

Moving to the TME stromal component, the negative prognostic role of the TC desmoplastic stromal reaction has been acknowledged for over a decade [30,31]: in this context, CAFs have more recently been highlighted as key players in the progression from DTC to PDTC [16,32,33]. CAFs exert several pro-tumoral functions: first, they interact with the extracellular matrix and thereby foster tissue fibrosis and desmoplasia; this results in a mechanical barrier, which acts against the development of an efficient host immune response. Interestingly, in TC the percentage of tissue fibrosis has been directly correlated to the degree of disease aggressiveness [34]. CAFs are also responsible for inducing chemokine interactions (CXCL12/CXCR4/CXCR7), increasing the secretion of metalloproteinases (MMP2 and MMP9), enhancing epithelial to mesenchymal transition (EMT) and activating numerous proliferative signaling pathways [16,35,36,37]. CAFs also increase the proliferation of monocytes and activated DCs, and promote the expression of several immune checkpoints (i.e., PD-L1, PD-L2, IDO-1, CTLA-4) [38]. In this respect, a higher concentration of CAFs in TC has been correlated to a higher lymph nodal metastasis risk, shorter overall survival and BRAFV600E genotype [37].

### 3.2. TME in TC: The Potential Role of Estrogens

A recent meta-analysis by Mannathazhathu et al. described the participation of estrogenic hormones in TC carcinogenesis, in line with the higher TC incidence—up to four fold—among females with respect to males. The authors included 19 studies (10 case–control and 9 cohort), from 1996 to 2017, assessing the association between reproductive risk factors and TC risk. A significant increased pooled risk for TC was reported in correlation with late age at menarche, increased parity, miscarriage/abortion and artificial menopause; conversely, the authors observed a protective effect of the prolonged use of oral contraceptives [39,40].

### 3.3. Estrogens and Thyroid Hormones

Estrogen and thyroid hormones (THs) are characterized by significant affinities and interactions: first, they share the same co-activators, including SRC-1, transcriptional intermediary factor 2 (TIF 2) and glucocorticoid receptor-interacting protein 1 (GRIP 1) [41,42,43]. Furthermore, THs activate the extracellular signal-regulated kinase ½ (ERK1/2) signaling by binding integrin and promoting proliferative cascades, while estrogens obtain this same effect through the SRC and PI3K/AKT signaling pathways. Moreover [42], the estrogen response elements (ERE) and THs-related elements (TRE), located in the specific regulatory regions of the target genes, share similar nucleotide sequences. As such, both estrogens and THs participate in hormone-dependent transcription [44].

At the hormonal level, T3 and E2 show reciprocal interactions: for instance, elevated levels of THs inhibit E2-dependent female sexual behavior [45]. Moreover, T3 regulates estrogen-induced gene expression while also promoting TGF-α production [44].

In summary, these findings suggest a competitive interaction between ERs and THs receptors (THRs), with a direct action on DNA structure, as well as an indirect impact on protein levels [42].

Open controversies remain surrounding the potential effect of phytoestrogens on several health-related endpoints: these compounds consist of estrogen-like molecules derived from plants, with isoflavones representing the most widely studied subtype. Phytoestrogens are mainly found in nuts and seeds, coffee, and in a variety of fruits and vegetables, including soybeans and processed products, alfalfa sprouts, garlic, celery, carrots, potatoes, apples and pomegranates, all of which are characterized by different phytoestrogens’ bioavailability. Of relevance, the average daily intake of phytoestrogens is much higher in Asian countries (20–50 mg per day) compared to Western areas (0.15–3 mg per day). As the structure of phytoestrogens is very similar to that of endogenous estradiol, these compounds are able to interact with ERα and ERβ. Therefore, questions have been raised concerning the potential role of phytoestrogens in hormone-dependent diseases, including certain cancer types [46]. Focusing on TC, a number of small, preclinical studies have investigated phytoestrogens’ potential properties in terms of antiproliferative effects and redifferentiation activity (i.e., their capability of re-inducing iodine retention in iodine-resistant TC) [47]. Notwithstanding a few in vivo studies that have been carried out in mice, a very limited number of human phase I/II clinical trials have been conducted in this field, specifically involving fosbretabulin: this microtubule destabilizing agent derived from the African bush willow has shown promising, albeit preliminary signs of activity against ATC when administered in combination with carboplatin and paclitaxel [48,49,50]. These initial data may promote further research in this fascinating and only partially explored setting.

### 3.4. Estrogen Receptors and TC

Estrogens promote cancer growth through both genomic and non-genomic trans-membrane ER-mediated pathways. Genomic effects are exerted through the two classical ERα and ERβ isoforms. Conversely, the non-canonical ER G protein-coupled estrogen receptor 1 (GPER 1) mediates non-genomic estrogenic effects [51]. In this regard, Bertoni et al. described a positive association between GPER 1 gene expression and mRNA levels of thyroid differentiation genes, with a lower GPER 1 expression correlating to more advanced TC stages and extra-thyroidal extension [52]. Despite ER levels in DTC being shown not to differ among female versus male patients, an overall higher ER expression has been associated with a higher Ki-67 and larger tumor size [53,54]. Moreover, while ERα seems to play a pivotal role in tumorigenesis, low ERβ expression has been correlated with poor prognosis in FTC [13].

More recently, estrogen-related receptor gamma (ERRγ) has also drawn researchers’ attention: ERRγ is an inducible transcription factor and a member of the NR3B estrogen-related receptors family. It is expressed across a variety of healthy tissues, where it participates in normal cellular development and homeostasis. Moreover, ERRγ’s high expression has been described across several malignancies including breast, prostate and gastric cancer and hepatocellular carcinoma [55,56,57,58]. In this respect, 40% of ATC and 60% of PDTC cases have shown ERRγ upregulation, which has led to the growing interest in light of its potential significance in the TC setting. As a result, therapeutic approaches involving ERRγ as a target are being explored, with sparse, preliminary encouraging data. In detail, DN200434 and GSK5182, two orally available ERRγ inverse agonists, were shown to increase sodium iodine symporter (NIS) trafficking and expression in ATC preclinical models, resulting in a synergic, anti-tumoral effect upon radioiodine therapy [59,60]. Further studies are required to better delineate the potential role of ERRγ, both as a TC signature biomarker, and as a future therapeutic target [61].

### 3.5. ER-Activated Pathways and TC

E2 has been demonstrated to significantly upgrade ERα expression in TC cell lines, promoting their proliferation through the activation of several transcriptional pathways: among them, the RAS/RAF/MAPK/ERK pathway has been found to be predominantly affected by estrogen-dependent stimulation in TC [62]. In this regard, epidermal growth factor receptor (EGFR) overexpression/mutation is a frequent, cancer-specific event in TC: in an Egyptian population (N = 60) with TC, over 50% of patients harbored EGFR mutations (deletion ex 19) [63]. Moreover, EGFR overexpression was documented in ATC, which is coherent with the preclinical activity of EGFR inhibitor gefitinib [64]. Of relevance, a recent metanalysis showed a correlation among EGFR extra-thyroid extension, nodes metastases and TNM stage [65]. The crosstalk between the EGF system and E2—consisting of a loop of reciprocal stimulation—has been sufficiently described: this may involve cytoplasmatic or membrane-associated receptor binding, which both rapidly activate intracellular signaling cascades, including ERK, PI3K and STATs [66]. Such molecular interdependence is likely to play a part in TC development and progression.

In addition, E2 modulates inflammation and apoptosis through different transcriptional factors. Notably, ERα and ERβ exert differential effects, as ERα inhibition and ERβ activation both result in the higher production of proliferator-activated receptor gamma (PPARγ), thereby inducing TNF-α, IL6 and NF-kB and promoting apoptosis^62^. Accordingly, ERα stimulation and ERβ inhibition were found to promote the survival and growth of PTC tumor cells in a dose-dependent manner, whereas ERα inhibition and ERβ stimulation led to the opposite effects. Interestingly, high ERα—and not Erβ—levels were found to correlate with the higher expression of hypoxia-induced factor 1 (HIF-1), the increase in downstream VEGF and rapid tumor growth in FTC and ATC [13].

Another key player in this setting is represented by the E26 transformation-specific family variant 5 (ETV5), an estrogen-mediated transcription factor involved in TC PI3KCA-dependent proliferation, migration and EMT. Although its precise molecular mechanisms remain unclear, ETV5 may regulate the cell cycle by influencing G1/S transition and inducing angiogenesis. Moreover, ETV5 modulates the PI3K pathway by inducing cell de-differentiation. Of relevance is that high levels of ETV5 expression have been associated with a negative prognostic impact in TC [67,68].

### 3.6. Estrogens and TME

The role of estrogen signaling in the modulation of the TME has been previously established across different tumor types [69,70]. Indeed, ERs are expressed across multiple immune cell populations, where they upgrade the extracellular wall expression of vimentin and metalloproteinases: this leads to a pro-tumoral, inflamed TME, characterized by the down-regulation of T effectors and NK cells’ expansion, and the promotion of angiogenesis and anti-apoptosis signaling [71]. In addition, in breast cancer, E2 was shown to inhibit NK- and T-cell-mediated tumor cell elimination through the impairment of granule-mediated exocytosis of serine proteases [72]. Moreover, Tai et al. demonstrated in vitro and in vivo Treg expansion after the administration of physiologic doses of E2 [73]. Figure 3 summarizes the main effects of estrogenic signaling on the TME in TC.

ER-mediated inflammatory, immune-suppressive action on the TME might also depend on ROS production and ROS-related pathways’ activation: indeed, estrogen/ERα have been shown to promote cell autophagy and ROS generation, resulting in positive feedback for tumor cell proliferation, survival and apoptosis. Moreover, ROS-mediated oxidative damage directly affects DNA structural integrity, increasing genomic instability and promoting cellular malignant transformation through a mutagenic, carcinogenic TME [74].

Concerning estrogens and inflammation, a recent study performed a bioinformatic analysis using The Cancer Genome Atlas (TCGA) and Gene Expression Omnibus (GEO) databases: the authors focused on the expression of estrogen-related genes in PTC, with a special highlight on Neuromedin U (NMU), a neuropeptide involved in several physiological and pathological inflammatory processes. Interestingly, the proliferative ability of PTC cells’ proliferation and KRAS pathway activation appeared significantly decreased after knockdown of the NMU gene, in line with the potential pro-tumoral, estrogenic activity on the TME within TC [75].

### 3.7. TC and Pregnancy

During gestation, THs are transferred to the fetus, as estrogens stimulate the production of maternal thyroxine-binding globulin (TBG). In this context, β-human chorionic gonadotropin (β-hCG) stimulates the maternal thyroid gland by cross-reacting with TSH-receptors. Therefore, maternal THs production curve follows that of β-hCG: that is, THs increase up to 50% during the first 5–6 months of gestation, then slowly decrease in between 12 and 16 weeks, dropping back to normal values throughout the second and third trimesters. Conversely, TSH levels gradually and consistently decrease throughout pregnancy [76].

The relationship between estradiol, β-hCG and TC has been addressed in the literature, as the high levels of β-hCG and estrogens along with the negative iodine balance during pregnancy are responsible for the growth of up to 30% of the thyroid gland: this entails the concurrent enlargement of preexisting thyroid nodules, as well as the formation of new nodules [77]. Several studies have supported a direct correlation between E2 levels and TC development (e.g., the inverse association between use of contraceptives and TC incidence [78,79]); for instance, Messuti et al. demonstrated higher persistence and recurrence of papillary thyroid microcarcinoma during pregnancy [80]. Conversely, other works have failed in confirming significant correlations with analogous factors (e.g., menarche or menopause, parity and bilateral oophorectomy [81,82]), leaving room for controversial conclusions. In this setting, our institution performed a retrospective analysis on 123 women who received a DTC diagnosis during pregnancy. In this population, ERα expression resulted as being higher in tumors that arose during pregnancy or in the first year after delivery, as compared with DTC that developed in women who were never pregnant or > 1 year after delivery (*p* = 0.01). Interestingly, DTC diagnosed during pregnancy was associated with a poorer prognosis compared to tumors developed in non-gestational time (*p* = 0.0001). Accordingly, at the stepwise logistic regression analysis, the diagnosis of DTC during pregnancy or in the first year postpartum appeared to be the most significant indicator of persistent disease (*p* = 0.001) [83].

### 3.8. TC and Metabolic Syndrome

As the ongoing increase in DTC incidence cannot be entirely explained by overdiagnosis, the parallel, dramatic rise in the prevalence of metabolic syndrome has raised doubts on a possible correlation between these two conditions. As thoroughly described in the literature, metabolic syndrome is characterized by higher estrogens: this is explained by increased fat mass, which leads to the higher expression of aromatase, a member of the cytochrome P450 superfamily that synthesizes estradiol and estrone through the aromatization of testosterone and androstenedione, respectively [84]. Moreover, leptin, which is primarily produced in white adipose tissue, is secreted in greater quantities in obese individuals, which also leads to a greater stimulation of aromatase activity [85]. Last, but not least, oxidative stress is also a player in metabolic syndrome, as ROS production leads to NF-κB activation and to increased transcription of pro-inflammatory cytokines, complement factors and matrix proteins [86]. All of the above-mentioned factors participate in the establishment of a low-grade, chronic inflammatory state, which is typical of metabolic syndrome, and which, in turn, lays the foundation for the development of a plethora of pathologic states—including cardiovascular, autoimmune, metabolic and, notably, oncologic diseases. As previously described, activated AKT/mTOR/PI3K and ERK /MAPK signaling fosters angiogenesis, cell proliferation, transformation and invasion, overall, promoting carcinogenesis [87].

In this context, several studies have assessed the potential impact of metabolic syndrome components on TC incidence and outcomes: interestingly, obesity, hyperglycemia (especially in female patients), hypertension and dyslipidemia do appear to contribute to TC development and progression; conversely, studies addressing the effects on TC prognosis have yielded more controversial results, owing to the lack of large-scale, long-term, prospective analyses [87].

## 4. Conclusions

The recent technical and medical advancements in the management of TC have led to the emergence of RAIR-DTC long survivors: as a result, the need for an updated therapeutic flow chart to further improve patients’ outcomes is becoming ever-so critical. Given such a scenario, a novel therapeutic armamentarium might be achievable through a deeper knowledge of TC pathogenesis and molecular mechanisms.

In the present review, we summarized the available evidence supporting the participation of estrogen-mediated signaling in TC natural history, with a special focus on the complex interactions between estrogens and the TME.

Based on the growing evidence of their intertwining role in TC development and progression, the TME and ER pathways could be considered as novel, potential therapeutic targets as, to date, the role of anti-estrogens and of selective ER regulators/down-regulators has not been clearly established. In this regard, combinations of different compounds (i.e., target therapies, redifferentiation drugs, immunotherapy) could further improve drug delivery and efficiency, fostering metabolic reprogramming, enhancing immune activation, and surmounting the pro-survival carcinogenic TC pathways. This is especially relevant for DTC, where TKI’s substantial toxicity profile poses specific clinical challenges. Indeed, the search for more conservative, personalized, risk-adapted treatment options is ongoing in this setting [88].

Prospective data are eagerly awaited, as the currently available literature, also including data on pregnant and other estrogen-exposed populations, appears largely heterogeneous and, as such, does not permit definite conclusions to be drawn.

## Figures and Tables

**Figure 1 cancers-15-02458-f001:**
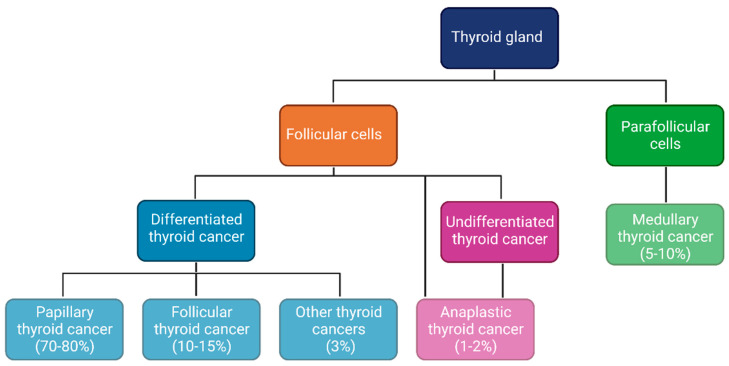
Thyroid cancer. Differentiated thyroid cancer (DTC) and poorly differentiated thyroid cancer (PDTC) derive from follicular cells. The most common form of DTC is papillary thyroid cancer (PTC), followed by follicular thyroid cancer (FTC). Anaplastic thyroid cancer (ATC) is the rarest subtype, deriving from PDTC or ex novo carcinogenesis of follicular cells. Medullary thyroid cancer (MTC) originates from parafollicular cells.

**Figure 2 cancers-15-02458-f002:**
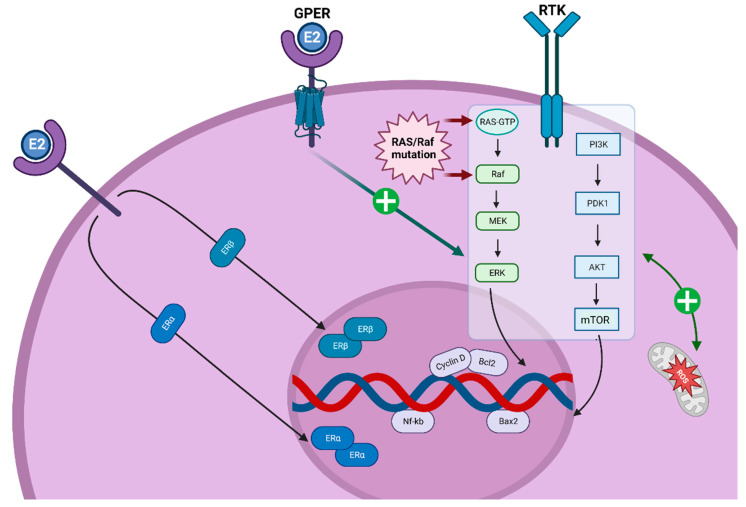
Estrogens in thyroid carcinogenesis. ER may play a role in thyroid carcinogenesis via genomic and non-genomic pathways, stimulating both the PI3K/AKT/mTOR and the RAS/Raf/MAPK pathways (as highlighted by the light blue box), and resulting in increased ROS production. These pathways are crucial in promoting cellular proliferation and differentiation. ER = estrogen receptor; GPER = G protein-coupled estrogen receptor; ROS = reactive oxygen species; RTK = receptor tyrosine kinase.

**Figure 3 cancers-15-02458-f003:**
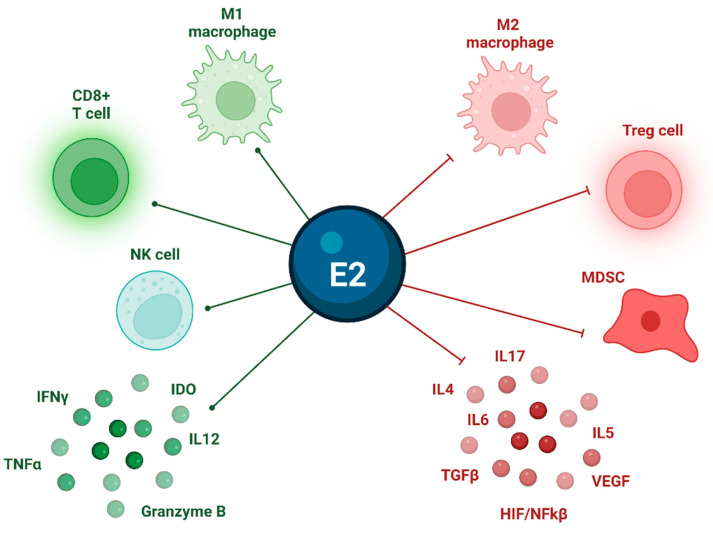
Effects of estrogenic signaling on TME in TC. E2 positive feedback (green line) induces M1 macrophages (anti-tumoral), CD8+ T cells, NK cells, and upregulates IFNγ, IL12, TNFα, Granzyme B; E2 negative feedback (red line) inhibits Tregs, MDSCs, M2 macrophages (pro-tumoral). E2 = estradiol; HIF/NFkβ = hypoxia-inducible factor/nuclear factor κβ; IDO = indoleamine 2,3-dioxygenase; IFNγ = interferon gamma; IL = interleukin; MDSC = myeloid-derived suppressor cell; NK = natural killer; TC = thyroid cancer; TGFβ = transforming growth factor beta; TME = tumor microenvironment; TNFα = tumor necrosis factor alpha; Tregs = T regulatory cells; VEGF = vascular endothelial growth factor.

**Table 1 cancers-15-02458-t001:** Ongoing studies on immunotherapy in TC.

Trial	Protocol	Expected Date
NCT04171622	Lenvatinib and Pembrolizumab for the Treatment of Stage IVB Locally Advanced and Unresectable or Stage IVC Metastatic Anaplastic Thyroid Cancer	August 2023
NCT04675710	Pembrolizumab, Dabrafenib, and Trametinib Before Surgery for the Treatment of BRAF-Mutated Anaplastic Thyroid Cancer	June 2024
NCT05059470	IMRT Followed by Pembrolizumab in the Adjuvant Setting in Anaplastic Cancer of the Thyroid (IMPAACT): Phase II Trial Adjuvant Pembrolizumab After IMRT in ATC	October 2023
NCT02628067	Study of Pembrolizumab (MK-3475) in Participants With Advanced Solid Tumors (MK-3475-158/KEYNOTE-158)	June 2026
NCT03360890	Pembrolizumab With Chemotherapy for Poorly Chemo-responsive Thyroid and Salivary Gland Tumors (iPRIME)	September 2024
NCT04061980	Encorafenib and Binimetinib With or Without Nivolumab in Treating Patients With Metastatic Radioiodine Refractory BRAFV600 Mutant Thyroid Cancer	June 2023
NCT05453799	Vudalimab for the Treatment of Locally Advanced or Metastatic Anaplastic Thyroid Cancer or Hurthle Cell Thyroid Cancer	July 2024
NCT05659186	PD-1 Inhibitor and Anlotinib Combined With Multimodal Radiotherapy in Recurrent or Metastatic Anaplastic Thyroid Cancer	December 2024

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
