# Peer review of "The Tumor Microenvironment and the Estrogen Loop in Thyroid Cancer"

_cancers, 2023, doi:10.3390/cancers15092458_

Round 1
Reviewer 1 Report
Denaro et al. investigated the cancerogenic role of estrogens in thyroid cancer, through activation of proliferative pathways and immunosuppressive effect on tumor microenvironment.
This is an interesting and original review. Few minor comments are suggested to improve on the submitted manuscript:
- The expression of EGFR is frequent in thyroid cancer and appears to be involved in the progression of papillary thyroid cancer. The authors should also speculate on the potential crosstalk between EGF system and estrogens in thyroid cancer. Please see and cite: PMID: 33435649, PMID: 18996136.
- The concept of personalized treatment for differentiated thyroid cancer should be briefly reported (please see and cite: PMID: 33213119), together with potential therapeutic applications in targeting the estrogen loop.
- This review is focused on the interactions between TME and the estrogen loop in thyroid cancer. I cannot find any reference regarding this in the paragraph “The impact of lenvatinib on TME”.
- Please check reference # 21, reported as “clinicaltrials.gov”. In the text: “Recently, a metaanalysis of Mannathazhathu et al. supported an association among reproductive hormones and risk of TC and explained female preponderance [21]”.
Author Response
We are thankful for the helpful comments.
We have applied the following changes:
- we added within our Results (2.4 ER-activated pathways and TC) a speculation on the potential crosstalk between EGF system and estrogens in thyroid cancer, and we thereby cited the suggested articles;
- we removed the paragraph “The impact of lenvatinib on TME” to improve the overall coherence of the manuscript;
- we corrected the reference # 21, and we reviewed all the references due to some previous mistakes for which we sincerely apologize.
Reviewer 2 Report
The present paper is well written.
In 3.2. TME: the emerging role of Estrogens As TC incidence is four time more common in females, estrogens may play a role in thyroid cancerogenesis [20].
Kindly note that you may cite the following book:
Tsatsakis A. (2021) Toxicological Risk Assessment and Multi-System Health Impacts from Exposure. Academic Press.
Author Response
We thank the Reviewer for the kind comment.
We have added the suggested citation as indicated by the Reviewer.
Reviewer 3 Report
This paper is supposed to be a review on the relationship between cellular estrogen receptors, circulating estrogens and tumor microenvironment (TME) in thyroid cancer. Unfortunately, the paper is a confuse sequence of paragraphs almost completely lacking a coherent line and of a conclusive message (“cut and paste” style?). An important contribution to this confusion derives from the almost complete mismatch between the references quoted in the text and the corresponding references listed in the “References” section. I have checked every single reference from 1 to 41 (i.e from Introduction to the end of paragraph 3.2.3), finding only 2-3 (possibly) correct! Of course, due to this diffuse mismatch, the sentences are often obscure and not supported by an adequate description of the pertinent experimental data.
In addition, I found other important criticisms, some of whom are reported below:
Background: No mention is made to the simple observation that although TC is more frequent in women, benign thyroid diseases (and in particular thyroid autoimmune disease) display even higher female preponderance. Moreover, poorly differentiated and anaplastic thyroid carcinomas show the lowest F/M ration among thyroid tumors.
Abstract. The abstract is unbalanced: 3/11 lines (about one third) are used to give detailed epidemiological data on TC, which are beyond the purpose of this review. On the other hand, the final message is vague and tautological (“thyroid microenvironment composition correlated with outcome”), without any logical link to the main topic of the review (estrogens and TC)
Fig 2. A Figure 2 as indicated in the Introduction (last line of page 2), supposed to summarize the genetic alterations involved in TC development, does not exist. Figure 2 is a confuse, poorly readable cartoon which should illustrate the “central role” played by estrogens in thyroid carcinogenesis. Actually, the figure only shows that, after binding to its ligand, ER may activate some pathways which are involved in carcinogenesis (not specific for TC). Moreover, ER is represented in the figure only as membrane ER (GPER-1) mediating rapid non-canonical effects. What about canonical ER? The pathways of thyroid hormone (TH) and TH receptors are also confusedly represented.
Paragraph 3.3 “The impact of Lenvatinib in TME” The entire paragraph seems unrelated to the main topic of the review (estrogens and TC).
Finally, no mention is made to the potential reciprocal relationship between estrogens, thyroid autoimmunity and thyroid cancer, a topic potentially strictly related to female gender and sex hormones.
Author Response
We apologize to the Reviewer for the mismatches within the References, which we have now entirely reviewed. We have also reviewed the writing style so that it is overall more fluent and we hope that the Reviewer will appreciate our efforts. In this respect, Dr. Rebecca Romanò helped in the review process, which is why she is now included as co-first author and corresponding author as an acknowledgement of her substantial contribution.
We addressed the important criticism that the Reviewer raised, specifically:
- we included the suggested epidemiological observations within our Background, also speculating on the potential reciprocal relationship among estrogens, thyroid autoimmunity and thyroid cancer;
- we reviewed the Abstract so that it is now more balanced, and it better reflects the content of the manuscript;
- we reviewed all the three Figures included in the manuscript using BioRender for a better final quality; in this regard, Figure 2 now includes both canonical and non-canonical ER-mediated pathways, and THs pathways were removed to obtain better coherence with the original aim of the Figure;
- we removed Paragraph 3.3 “The impact of Lenvatinib in TME”, to improve the overall coherence of the manuscript.
Round 2
Reviewer 3 Report
The paper has been fully re-written and now is a clear, comprehensive and updated review on the potential role of estrogens and estrogen receptors in thyroid carcinogenesis. Figures were also substantially improved and all the references are now correct.
Regarding the complex relation between thyroid cancer and thyroid autoimmunity, I am wondering whether Authors may consider the quotation of the following recent and comprehensive review published in Cancers: Pani et al. The Immune Landscape of Papillary Thyroid Cancer in the Context of Autoimmune Thyroiditis. Cancers 2022, 14, 4287.https://doi.org/10.3390/cancers14174287, which provides an extensive review of the clinical and experimental studies in this field.
Author Response
We thank the Reviewer for the kind comments and we have now added among the references the above suggested work concerning the complex relation between thyroid cancer and thyroid autoimmunity.
We believe our manuscript should be now ready to be finally published.
We thank you again and we remind you our authorship changes, where Dr. Rebecca Romanò has been included as co-first and co-corresponding author as an appreciation to her valuable contribution to this final, improved version of the manuscript. We thank again the Reviewers for their precious contribute which allowed such improvements. Please do not hesitate to contact us for any further doubts or comments.